# Elements of Pathway for Quick and Reliable Health Monitoring of Concrete Behavior in Cable Post-Tensioned Concrete Girders

**DOI:** 10.3390/ma14061503

**Published:** 2021-03-18

**Authors:** Lukasz Bednarz, Dariusz Bajno, Zygmunt Matkowski, Izabela Skrzypczak, Agnieszka Leśniak

**Affiliations:** 1Faculty of Civil Engineering, Wroclaw University of Science and Technology, Wybrzeże Wyspiańskiego 27, 50-370 Wrocław, Poland; zygmunt.matkowski@pwr.edu.pl; 2Faculty of Civil and Environmental Engineering and Architecture, UTP University of Science and Technology, Al. prof. S. Kaliskiego 7, 85-796 Bydgoszcz, Poland; dariusz.bajno@utp.edu.pl; 3Faculty of Civil and Environmental Engineering and Architecture, Rzeszow University of Technology, Powstanców Warszawy 12, 35-082 Rzeszow, Poland; izas@prz.edu.pl; 4Faculty of Civil Engineering, Cracow University of Technology, Warszawska 24, 31-155 Kraków, Poland; alesniak@l7.pk.edu.pl

**Keywords:** post-tension, cable, concrete, girder, diagnostic, destructive test, non-destructive test, structural health monitoring, safety

## Abstract

The paper discusses the problems connected with long-term exploitation of reinforced concrete post-tensioned girders. The scale of problems in the world related to the number of cable post-tensioned concrete girders built in the 1950s and still in operation is very large and possibly has very serious consequences. The paper presents an analysis and evaluation of the results of measurements of the deflection and strength and homogeneity of concrete in cable–concrete roof girders of selected industrial halls located in Poland, exploited for over 50 years. On the basis of the results of displacement monitoring in the years 2009–2020, the maximum increments of deflection of the analyzed girders were determined. Non-destructive, destructive, and indirect evaluation methods were used to determine the compressive strength of concrete. Within the framework of the indirect method recommended in standard PN-EN 13791, a procedure was proposed by the authors to modify the so-called base curve for determining compressive strength. Due to the age of the analyzed structural elements, a correction factor for the age of concrete was taken into account in the strength assessment. The typical value of the characteristic compressive strength is within the range 20.3–28.4 MPa. As a result of the conducted tests, the concrete class assumed in the design was not confirmed, and its classification depended on the applied test method. The analyzed girders, in spite of their long-term exploitation, can be still used for years on the condition that regular periodical inspections of their technical condition are carried out. The authors emphasize the necessity for a permanent and cyclic diagnostic process and monitoring of the geometry of girders, as they are expected to operate much longer than was assumed by their designers.

## 1. Introduction

In many cases in construction practice, it may not be possible to carry out a precise assessment of the technical condition of buildings and their components using traditional, commonly used methods. Such situations will concern buildings and structures with a large number of structures very sensitive to environmental conditions, for which the assumed technical life has already significantly exceeded their current service life. It will not be practicable to replace these elements with new ones due to the large number of buildings constructed with such and similar technologies. An example of such facilities are industrial halls built using prefabricated elements made in prestressed concrete technology.

Post-tensioning techniques have been widely applied in construction to design bridge piers and decks, building slabs, and long-span girders [1]. Post-Tensioned structural elements are very often used in bridges i roof construction industrial halls due to their ability to economically span long widths while providing an aesthetically structure [2,3]. Due to these advantages and effectiveness of the structural behavior post-tensioned pre-stressed concrete girders have been constructed for the last half century, [4]. Cable post-tensioned concrete girders are reinforced concrete constructions which are preloaded with prestressing forces either after the concrete has partially or completely hardened in the molds, or directly on site with the use of cables, anchored at the ends of the elements [5,6]. The cables (prestressing tendons) are inserted into the channels mainly located inside, so that the strength of the element can be used rationally in each section.

Tension presses (jacks) are used to prestress the structure. The space between the tendons and the walls of the cable tunnel is filled with an injection material (e.g., cement–water grout) of sufficient strength to protect the steel of the tendons against corrosion and to transfer the prestressing forces to the structure.

With a few exceptions, these structures have a segmental structure, joined directly on site during assembly by means of welded joints, filled with mortar. In spite of the impression made by their seemingly massive external appearance, these elements are quite “delicate” and have a long service life (approx. 50–75 years), hence the serious problem of their safe use; i.e., maintenance in an appropriate technical condition has already become a topic of discussion [2,7,8,9].

The beginnings of tests on prestressed reinforced concrete structures date to the second half of the 20th century [2]. Research was conducted both in Europe and in the United States. The introduction of this new type of reinforced concrete structure was intended to eliminate the disadvantages of concrete and reinforced concrete, which was characterized by very low tensile strength and high susceptibility to cracking. The lack of materials with the required properties meant that these structures had not been used before. The development of high-strength steel for prestressing tendons and the possibility to repair war damage more quickly (after World War II) brought about a change, mainly in industry, for the piercing of long-span halls. In 1952, the International Association of Prestressed Structures was founded, which in 1988 became part of the International Concrete Association. The first industrial-scale use of prestressed concrete began in the 1950s, mainly for the manufacture of prefabricated girders, roof and floor slabs, and railway sleepers [10].

There are not many studies in the literature of specific standard guidelines for the testing and maintenance of these girders apart from a few technical manuals or handbooks, e.g., [11,12,13,14,15]. Current knowledge on the serviceability of cable post-tensioned concrete girders is based on the assessment of their technical condition. The perspective of maintaining building objects in an adequate technical condition is one of the most important problems during their service life [16]. Mostly visual inspection, non-destructive testing, measurements, calculations, and when necessary, in-depth material tests are carried out for the diagnosis of technical conditions. The methods of material testing of structures are quite well known and are described in the literature [17,18,19,20,21]. Depending on the degree of their invasiveness, these can be divided into destructive, semi-destructive, and non-destructive methods [22]. Unfortunately, some of the material destructive tests (DT) may not be feasible due to the possibility of damaging the tested cable post-tensioned concrete structures [23,24]. Aging of materials, particularly of concrete, as well as the influence of concrete aging mechanisms such as ASR (alkali–silica reaction), DEF (delayed ettringite formation) or ACR (alkali–carbonate reaction) are separate issues.

Hence, the need has arisen to propose an appropriate and reliable methodology for testing materials via non-destructive testing (NDT) [25,26,27,28,29,30,31]. Specialized NDT tests can only be performed by a limited number of main research centres (laboratories and accredited institutes), and the scope of the need is enormous, because the period 1960–1980 saw growth in this type of design.

Monitoring the long-term behaviour of civil structures is of great importance for damage prevention and design improvement [32]. Technical condition assessments carried out on the basis of continuous monitoring of a structure make it possible to control the ultimate and serviceability limit states and thus the risk of possible overloading. Structural monitoring may be defined as all techniques and methods aimed at measuring, among other things, the deformation of structural elements as a function of time. Geometric monitoring techniques usually involve measurement of deformation and displacement using various methods, including lasers [33].

The application of an appropriate approach to the monitoring of structures, as proposed, e.g., in [34,35] and in particular, the monitoring of cable post-tensioned concrete girders [33,36] as well as NDT material testing and repair of such elements [37] as a long-term solution, may lead to benefits that cannot be achieved otherwise, saving time and avoiding costs. Attention may also be drawn to the increasing spread of new, innovative materials for the protection and strengthening of concrete structures, such as nanomaterials [38,39] or composite materials [40,41,42]. In addition, an appropriate approach to repair may help avoid negative environmental impacts [43,44], service interruptions, overloading of the nearby infrastructure, etc. An additional feature guaranteeing the usefulness of structure monitoring is non-invasive, constantly updating information about its condition and the possibility to react in advance, which increases its safety and durability and thus extends its service life.

This paper presents selected problems of the evaluation of the technical condition of cable post-tensioned concrete girders. In particular, the path of their quick monitoring is presented. The results of deflection measurements and tests of compression strength and uniformity of concrete were evaluated. The subject of consideration was selected roof girders of two industrial halls located in southern Poland.

## 2. Materials and Methods

### 2.1. Description of Seleceted Cable Post-Tensioned Concrete Girders

The structures proposed for the case study are divided into two groups: typified and non-typified (individually designed) cable post-tensioned concrete girders (type: KBO—monolithic one-piece girders and KBOS and KBS—girders assembled on the building site from prefabricated elements) with spans ranging from 15 to 27 m. Their shape is an arched upper chord, “broken” into ~3 m long sections, and a straight lower chord (tie beam), which are connected by vertical posts (Figure 1 and Figure 2). The top chord of these girders has a parabolic shape, while the bottom chord has a rectilinear shape. The prestressing cables 12 Ø5, the number of which depends on the type of girder, are routed in the inner channels of the bottom chord.

The subject of the study are the girders built-in in two halls located in southern Poland: Hall A—17 girders with a span of 27 m (girder type KBS/27—Figure 1 and Figure 2c), Hall B—23 girders with a span of 24 m (girder type KBOS/24—Figure 1 and Figure 2a). The post-tensioned concrete girders used in Halls A and B are located in an environment with a weak (low) degree of aggressiveness and class “B” susceptibility to corrosion hazards; therefore, they are subject to obligatory periodical technical inspections, which should be carried out at intervals of no longer than 18 months. In [45], it is stated that if any irregularities should occur in the external appearance of these structures, a change in their condition of use or in the type and amount of loads, technical inspections should be carried out once or at intervals of no longer than five years. Under the regulations applicable in Poland, building structures with a surface area of more than 1000 m^2^ or a roof area of more than 2000 m^2^ require inspection twice a year.

Prefabricated ribbed roof slabs are usually supported on the top chords of the girders for hall buildings. The original roof covering, laid directly on the roof plates, consisted of at least three layers of roofing felt (it had been repaired many times), two layers of insulating fiberboards with a total thickness of approx. 2.5 cm, and a levelling layer of cement mortar with a thickness of approx. 1.5–2.0 cm. Subsequent thermo-modernization was carried out on the existing roof covering, which also increased the permanent load on the roof. This modernization consisted in laying hard mineral wool up to 20 cm thick and covering it with two layers of weldable roofing paper. Installations (lighting, ventilation, heating) were suspended from the lower girder strips, and roof windows (skylights) with a steel structure and single reinforced glass filling (Figure 2c) were rebuilt over the central strip of Hall A.

### 2.2. Diagnostic Methods Description

Diagnostic tests allow evaluation of the condition of reinforced concrete prestressed girder structures, and the degree of deterioration of individual elements, and determination of the characteristics of concrete and reinforcement, and the causes of damage, and prediction of the durability of the structure [46]. In the diagnostic testing of building structures or their elements, the actual condition and the actual conditions of their operation should be taken into account, and current standards should be used in the assessment. Former standards that were the basis for the design of these structures should be treated as technical knowledge [45]. As far as the types of diagnostics are concerned, we can mention: periodic diagnostics (required by the operation of the object); ad hoc diagnostics (after finding significant irregularities in the work of elements); and full diagnostics (in connection with the planned modernization of the object). General principles of diagnostics of existing prestressed girder roof structures list the following activities [11]: visual inspection of girders and damage inventory, examination of the degree of filling of cable channels, examination of the distribution and quality of reinforcement, geodetic examination of girder deflections, macroscopic examination of concrete, imparting of concrete strength and uniformity, corrosion examination of concrete, and evaluation of cable corrosion hazard. Testing should include [11]: visual observation and assessment, “in situ” testing of structures or their components, and laboratory testing.

In this paper, the authors will present selected measures, including: visual inspection of girders (visual method), geodetic investigation of girder deflections, and selected concrete tests. These are elements of the developed simple and effective monitoring of the state of the girders, which the authors have been using for over ten years. It is a method of cyclic, precise updating of the measured deformations combined with macroscopic examination of all elements. In particular, the examination of the quality and class of concrete connections of prefabricated elements and the extent of corrosion according to the pathway are presented in Figure 3. Material and static deformation analysis is carried out at the same time of the year, under similar recorded climatic conditions taking into account the level of protection of structures susceptible to thermal movement (e.g., is there insulation, and, if so, what kind?).

The use of a proven and reliable fast-track procedure (including material analysis) is necessary when a large number of girders are involved in a single structure or assembly (there can be up to 150 girders in a single object). The tests performed should be supported by an appropriate computational algorithm. From a financial perspective, this may not be cost-effective unless it is done automatically or when changes are significant. This is especially the case if you use an already known algorithm of calculation.

The ultimate limit state is the design and check for the safety of a structure and its users by limiting the stress that materials experience. It would be difficult to determine these values in a simple (measured) way without performing verifiable strength calculations, analyzing the actual static schemes of the structure as well as its verticality, and the flatness and deformation of the girders (taking into account imperfections) and their current properties and parameters, in this case, mainly concrete. The assessment of the serviceability limit state of buildings in intensive use and their structures consists, among others, in comparing the actual values of deformation of their geometry with the permissible values in relation to previous testing seasons. For this purpose, generally available measuring instruments or devices are used.

Technical inspections should be carried out immediately any changes in the appearance of even a single girder are noticed. Additional load on the roof (addition of approx. 0.4 kN/m^2^ on the whole surface of the roof) as part of thermomodernization work of hall B and in the case of hall A replacement of the carpentry of the roof windows (subtraction of approx. 0.5 kN/m on the whole surface of the skylights) already represent a requirement to check the capacity of the girder.

The proposed pathway for quick and reliable health monitoring of material behavior in cable post-tensioned concrete girders (Figure 3) is very helpful. These types of structures are resilient, yet sensitive to environmental conditions and aging processes, especially if they have already exceeded their service life. The authors of this paper think that further abandonment of cable post-tensioned concrete girders in their current location will be possible only if the procedures described in the above-mentioned algorithm are followed, which will make it possible to ensure their appropriate level of reliability. An essential element of this process will be to successively carry out reliable material tests, the results of which will decide the fate of these structures and that of the entire facility.

### 2.3. Visual Method

A visual inspection of the girders includes a preliminary assessment of the structural condition. During a visual inspection carried out as part of the periodic technical inspection, particular attention should be paid to:cracking and deformation of prestressed structures in monolithic sections and assembly joints, as well as the state of supports,defects in the concrete in the prestressed structures themselves and in the elements joined to them as well as the corrosion of the connecting elements,scratches and deformation of roof slabs, especially on their undersides, near the ends of girders, as well as roof leaks, with particular attention paid to the location of cable anchorage zones,the accumulation of dirt (dust) on the elements and the extent and weight of suspended installations (including: the change of their location and weight, replacement, or extension), information boards and other loads,cracks and defects in the glazing of skylights or replacement of their constituent elements with heavier ones,use of the building in accordance with its original purpose,thermal and humidity parameters of the building envelope and thermal and humidity conditions of the internal environment,possible influence of dynamic loads on the whole structure of the building, e.g., connected with the production technology of the plant.

In the case of regular inspection by visual methods, e.g., by UAV, it is possible to use the very accurate and fast DIC (Digital Image Correlation) optical method for the analysis of the state of preservation of the material (discussed in more detail, e.g., in [47,48,49,50]).

### 2.4. Deflection Measurements

Systematic measurement of girder deflections, using precision levelling, makes it possible to assess the correctness of girder behavior and can signal possible failure risks. One should always strive to make deflection measurements under comparable conditions. Maximum allowable deflection and displacement values should be taken according to the recommendations of investors, supported by standard requirements, e.g., [51]. In general, for reinforced and prestressed concrete structures, these should not exceed 1/250 of their theoretical span, assuming that the permanent loads on flat roofs vary from 30 to 50% (thus, the variable loads are in the range of 50 to 70%). Under the assumptions given above, it should be noted that in extreme cases, the maximum allowable deflection should not exceed 75 mm for a 27 m span girder (67 mm for a 24 m span member).

Based on the permissible vertical deflection values, it should be considered with what accuracy they can be reliably determined in reality. It seems that an accuracy of 10% of the measured deflections is sufficient from the point of view of the user and the small cost of the measuring equipment. Therefore, for a 10 mm deflection, the accuracy of the measuring equipment should not be more than ±1.0 mm.

It is important to bear in mind that the strain measuring system should be able to withstand external environmental factors such as strong gusts of wind, rolling or dropping loads, etc. during the measurement. In addition, periodic measurements of deformation should be carried out in similar outdoor as well as indoor climatic conditions. The influence of temperature on the operation of the structure must not be neglected and must be taken into account in the final measurement results.

Commonly known displacement measurement methods have their limitations, of which one should be aware before the type of monitoring is chosen. The method of measuring vertical displacement relative to the floor or stop crane platform should be immune to measurement interference, and resistant to measurement interference, which can be difficult in buildings with heavy traffic or production. Measurements would then have to be taken during periods of production stoppage. This method of horizontal measurement under the roof does not affect the use of the building. Detailed possibilities for such a measurement methodology are presented in [52].

Measurements of the amount of deflection can indicate those that show excessive disturbing deformation. In the given questionable cable post-tensioned concrete girders, the state of preservation of the material and its parameters should be thoroughly checked (using the NDT method). The complete measurements and tests are used for the overall assessment of the elements and to assign them to a specific class:I—good condition,II—acceptable condition,III—poor condition,IV—very bad condition.

Classification of the technical condition of the girder/girders to classes III and IV necessitates conducting extensive technical and material tests and carrying out urgent repair recommendations. For the technical condition of an object/element classified to classes I and II, on-going maintenance work and subsequent inspections in the current cycle (according to the established schedule) are required, as well as repair and removal of defects.

### 2.5. Concrete Testing Methods

The compressive strength of concrete is a basic parameter which proves its quality and durability. Achieving the desired quality of concrete involves appropriate design of concrete mixes [53,54], proper manufacturing [55], development of innovative research methods that aid concrete design aimed at obtaining appropriate properties and durability [56], and development of methods for analyzing assessment results both during production and in existing constructions [57,58].

The evaluation of concrete strength in existing building structures can be carried out in accordance with standard PN-EN-13791 [59], which organizes the rules for evaluating the compressive strength of concrete structures and precast concrete products. The method of determining the characteristic value of the strength of concrete in structures based on core boring can be determined in accordance with PN-EN 13791 [59], which is based on the recommendations of PN-EN 206 [60], thus determining the properties of concrete based on compatibility criteria.

The standard recommendations for strength assessment in structures according to PN-EN-13791 [59] concern three cases:Demonstration of compliance of the compressive strength of concrete in a structure, that is, concrete already embedded in the structural element (e.g., precast concrete), but without the use of “standard” samples. The indirect methods are used here, that is, those that do not “destroy” the structure, and at the same time are cheaper than traditional sampling.Evaluation of the quality of concrete in the case of non-compliance of the compression strength conditions, which was carried out with the use of standard samples or when execution errors are found during the execution of work.Evaluation of the technical condition of existing structures when they are to be modernized or redesigned.

The PN-EN 13791 standard [59] provides formulas for the analysis of test results, while the performance of the tests themselves is described in the four-tool standards of the PN-EN 12504 series [61,62,63,64]. They concern core drilling and sclerometric pull-out and ultrasonic testing. The compressive strength of concrete in structures is evaluated by non-destructive and/or destructive methods.

#### 2.5.1. Non-Destructive Methods—Sclerometric and Ultrasonic Tests

The development of non-destructive testing (NDT) of materials in civil engineering is mainly concerned with the detection of flaws and defects in concrete elements and structures [22]. In indirect methods, values other than strength are measured. To obtain reliable information about the compressive strength of concrete, it is necessary to calibrate the adopted test method proposed in PN-EN 13791 [59]. The standard provides for the use of indirect methods (NDT), in which the following quantities are measured:rebound number (sclerometric method)—PN-EN 12504-2 [62],pull-out force (“pull-out” method)—PN-EN 12504-3 [63],ultrasonic wave speed (ultrasonic method)—PN-EN 12504-4 [64].

In practice, many different methods using different measuring instruments are used. Characteristics of NDT methods of concrete are presented in Table 1.

In the case study presented in this paper, sclerometric and ultrasonic tests were used. These methods are used often and successfully to evaluate the quality of concrete in existing structures [65,66]. The accuracy of estimating the strength of concrete with sclerometric and ultrasonic methods is about 15–20%, while with the drilling method accuracy it is about 10%. When using all these methods in one element, the accuracy increases and amounts to 5–10% [67].

#### 2.5.2. Destructive Methods—Concrete Cores Test

In the case of the direct method (destructive method/DT), tests are carried out on samples cut out of the structure through core drilling. A description of the testing is given in the tooling standard PN-EN 12504 series [61] concerning core drilling. These boreholes should have a diameter from 50 to 150 mm, while those with a diameter from 100 to 150 mm will allow the use of the direct correlation relation obtained in the strength test in relation to the tests carried out on standard specimens, while diameters less than 100 mm require the use of correction factors. Correlation relationships relating to specimen dimensions, quantity, appearance, and the influence of these factors on the interpretation of the results obtained are given in PN-EN 13791 [59].

Specimens should be drilled in the structure, from those places where section weakening is least likely to affect the load-bearing capacity of the structure, then tested in the laboratory. Nondestructive testing should cover from 2% (for good homogeneity) to 100% (for poor homogeneity) of the girders in a given structure [45]. Examples of the application of the “direct” method can be found in the literature [68,69].

## 3. Results

### 3.1. Visual Method Results

All analyzed girders were visually inspected for structural and material damage using a visual method (using an elevator or more easily and quickly using a 4K camera with a resolution of 4000 × 3000 pixels, mounted on a DJI Inspire 2 unmanned aerial vehicle (UAV) with a gimbal (Figure 4).

Examples of cable post-tensioned concrete girder damage in Hall A identified as a result of the inspection shown in Figure 4.

During the operation of Hall A, damage occurred to the lower flange of the girder when the oversized structure was moved. The workers either did not notice this or ignored it. This coincided with the significant addition of loads during the replacement of a window skylight. The deflections of this girder were observed in the measurement results. At present, it is not known at what exact time (within one year) the damage shown in Figure 5 occurred. As can be observed, the damage is significant, the crack size is on the centimeter level (Figure 5a,d,e), the girder is twisted (Figure 5b,f), and there is significant damage of the concrete (Figure 5c). No cracks were observed on the other elements that did not meet the conditions of the standard [51], i.e., at exposure class X0 greater than 0.2 mm. Considering the non-aggressive exposure class, these values were not considered dangerous.

Was this a direct result of the exceptional load applied or did it propagate over time? Such situations may occur during the continuous operation of a production plant. During the following periodic inspection of the hall, the danger presented by a damaged bottom chord became apparent. Cracks, delamination, and deformation of the lower prestressed chord had completely changed its static scheme. Instead of compressing the concrete section, the tension of the cable caused it to be compressed by a large eccentric force that extended beyond the core of the chord section. It was only by chance that the damage to one girder, whose lower flange had lost its load-bearing capacity, did not lead to a widespread building disaster. After this event, two adjacent girders were carefully observed, for which deformation diagrams are shown in Figures 6–8.

### 3.2. Deflection Measurements

In the analyzed Halls A and B, measurements of girder deflections have been carried out since 2009. The measurements were performed with the trigonometric levelling method, using an electronic total station (Leica TCRP 1203 R100). The height differences between three points on each cable post-tensioned concrete girder were measured. The points were marked with markers on the lower flanges of the concrete girders, two at the ends and one in the middle of the span. Measurements were taken with comparable accuracy ±1 mm. The reference level for measurements was the altitude level of the hall floor, the fixed benchmark of the national PL-KRON86-NH network with the value of + 105.50 m above sea level for Hall A, and +114.10 m above sea level for Hall B. Girder maximum deflection increments Δ*f* in the middle of the span, monitored from 2009 to 2020 in Hall A was 25 mm, and in Hall B 32 mm. The results are summarized in Figure 6, Figure 7 and Figure 8 for Hall A and Figure 9, Figure 10 and Figure 11 for Hall B.

### 3.3. Sclerometric Test Results

Current procedures for testing reinforced and prestressed concrete structures using the sclerometric method are included in PN-EN 12504 [62]. According to the introductory note in the standard [62], the reflection number determined by the sclerometric method can be used to assess the homogeneity of concrete in a structure and to determine areas and sections of the structure where the concrete is of poor quality or where its quality has deteriorated. However, it is made clear that the test method is not considered an alternative to the determination of the compressive strength of concrete [70], but with the use of proper correlation, it can allow the estimation of the strength of a structure [59]. Thus, the base curve provided in EN 13791 [59] cannot be used in the assessment of concrete strength without appropriate scaling on boreholes taken from the structure.

For this reason, the recommendations of B-06250 [71], which allows non-destructive methods to evaluate the strength and quality of concrete, were used in the evaluation of both compressive strength and quality.

Testing of four roof girders was carried out with a Schmidt N-type sclerometer. In each structural element, measurements were made on lateral (vertical) surfaces at 15 measuring points. The scaling curve was determined by selecting a hypothetical scaling curve on the basis of reflection number tests and compressive strength (determined by the failure method). As a criterion for the selection of the hypothetical scaling curve, the condition was adopted that the value of the mean relative square deviation was *ν_k_* < 12%. The hypothetical scaling curve was assumed in the form of Equation (1):
*f_c_* = 0.0418 × *L*^2^ − 0.932 × *L* + 7.5 (MPa)(1)
where:*f_c_*—compressive strength of concrete,*L*—number of rebound.

For this equation, the value *ν_k_* = 23.6% > 12% was too high. Therefore, correction factor c was introduced, calculated as the ratio of the average strength of the samples taken in the press (*f_c mean_*) to the average strength of the specimens determined from the hypothetical scaling curve (*f_c,h mean_*). This ratio was *c = f_c mean_:f_c,h mean_* = 1.15. The corrected scaling curve was therefore of the form of Equation (2).
*f_c_* = 1.15 × (0.0418 × *L*^2^ − 0.932 × *L* + 7.5) (MPa)(2)

For this curve, *ν_k_* = 10.5% < 12%. This curve could be used to determine the compressive strength of the concrete based on the reflection number *L*. The log of sclerometric measurements for one prestressed element (G11 from Hall A) is shown in Table 2.

The characteristic compressive strength can be determined with formulas defined in [71]:1st criterion: f_i,min_ ≥ f_ck cube_,2nd criterion: *f_cm_*
*≥* 1,2 *f_ck cube_,*3rd criterion: *f_cm_*
*≥ f_ck cube_ +* 1.64·*s*

Thus, according to [71], the minimum value from the values should be taken as the concrete class identified with the characteristic compressive strength of concrete:*f_ck_* ≤ 32.4 MPa*f_ck_* ≤ 36.0 MPa/1.2 = 30.0 MPa*f_ck_* ≤ 36.0 MPa—1.64·1.92 MPa = 32.8 MPa.

Based on sclerometric tests, the characteristic compressive strength value is 30.0 MPa. Using the recommendations from [71,72], the homogeneity coefficient of concrete can be calculated using Equation (3):(3)$$k=1-1.645\;\times v_{k}$$

Based on the obtained value of homogeneity, the coefficient is *k* = 0.91 and relative standard deviation $v_{k}=5.1\%$. According to [72], the concrete quality for the selected element, girder G11 in Hall A, is very good.

The homogeneity of hardened concrete from sclerometer tests for the four girders was determined and the results are given in Table 3.

Both the estimated characteristic compressive strength of 30.0 MPa for girder G11 in Hall A and the assessed concrete quality based on homogeneity factor *k =* 0.91 was very good and the quality of other considered beams (Table 3) seem to be overestimated values; therefore, other methods of determining the compressive strength of concrete, such as ultrasonic or borehole methods, should be used.

### 3.4. Ultrasonic Tests Results

Ultrasonic tests were used as an alternative non-destructive method to estimate the characteristic compressive strength and to evaluate the homogeneity of concrete, which is allowed by PN-B 06250 [71].

Ultrasonic tests were carried out according to the principles given in PN-EN12504-4 [64]. A CT1 m with heads at a frequency of 100 kHz was used in the ultrasonic tests. The velocity was measured using the so-called crossed and semi-direct methods (Figure 12); the number of measurement sites per element was 15.

The layout of the measurement bases is shown in Figure 13.

The scaling curve was determined by selecting a hypothetical scaling curve analogous to the ultrasonic method. The final corrected scaling curve was in the form of Equation (4).
*f_c_* = 4.125 × *v_L_*^2^ − 12.18 × *v_L_* + 7.2 (MPa)(4)
where:*v_L_*—ultrasound wave speed (km/s).

A detailed example log of the ultrasonic tests carried out for one of the girders (G11 from Hall A) is summarized in Table 4.

Based on the results of ultrasonic tests, the characteristic compressive strength of concrete can be estimated using the recommendations of standard [71]. According to [71], the minimum value from the values should be taken as the characteristic compressive strength of concrete:*f_ck_* ≤ 30.2 MPa*f_ck_ ≤* 34.2 MPa/1.2 = 28.5 MPa*f_ck_ ≤* 34.2 MPa—1.64·2.11 MPa = 30.7 MPa.

Based on ultrasonic tests, the characteristic compressive strength is 28.5 MPa.

Using the recommendations of [73], the homogeneity coefficient can be calculated using Equation (5):(5)$$k=1-1.645\times v_{k}$$

Based on the obtained value of homogeneity coefficient *k* = 0.90 and relative standard deviation $v_{k}=6.2\%$, according to [73] the concrete quality for the selected element: girder G11 in Hall A, is very good.

The homogeneity of hardened concrete from ultrasonic tests for the four girders was determined from the recommendations in [73,74]. The results are given in Table 5.

Both the estimated characteristic compressive strength of 28.5 MPa for girder G11 in Hall A and the evaluated concrete quality based on uniformity coefficient *k =* 0.90 were very good and the quality of other considered beams (Table 5) seem to be values divergent from the values obtained in the sclerometer test; therefore, the other method recommended in [59] using correlation relationships of destructive and nondestructive methods and the test based method recommended in [75] should be used.

### 3.5. Concrete Core Tests Results

Strength tests of concrete cores, cut directly from existing building structures, are considered to be the most reliable. This situation changed significantly since the publication of the standard [59], which clarified the rules for assessing the compressive strength of concrete in structures and precast concrete products. The standard PN-EN 12504-1 [61] organized the procedures related to the collection and preparation of samples cut from core borings for testing. In our case, a Hilti rig with Ø100 mm and Ø150 mm core drills was used.

United Kingdom recommendations [76] suggest that one should aim to take cores with the smallest possible ratio of h/d *=* 1.0–1.2 due to the low cost of drilling, structural repairs associated with testing, and variability of concrete properties along the elevation and the influence of “scale effect”. In the case in question, h/d *=* 1.0 and the value of the scale factor was assumed to be 1.0 [76].

An important issue in destructive testing is the determination of the minimum number of specimens to be tested. PN-EN 13791 [59] treats this issue very generally and emphasizes that from the statistical point of view and with respect to safety requirements, it is recommended to use as many boreholes as practicable to evaluate the compressive strength of concrete in the structure. In the case under study, two samples each were taken (in the support zone of the beams) for four selected beams, for a total of eight core samples.

The concrete samples taken by means of cores were used to determine the actual compressive strength of concrete in a destructive manner. The samples were prepared and then tested in a destructive method in a strength press. The compressive strengths of the concrete are presented in Table 6.

The method recommended in [59] was used to determine the compressive strength based on a correlation relationship determined using a limited number of core borehole test results (number of samples *n* < 18) and sclerometric test results.

To determine the correlation relationship between the compressive strength of concrete in the structure and the indirect test result, the correlation relationship of the strength obtained for cores drilled from the structure and the test results performed for nondestructive testing must be determined, thus obtaining pairs of test results. Determination of the correlation relationship involves fitting a straight line or curve via regression analysis of pairs of results obtained from the test program. The indirect method measurement result is considered a variable value and the determined compressive strength of concrete in the structure is a function of the variable. The standard [59] emphasizes that the correlation relationship is determined assuming the possibility of a ten per cent underestimation of strength. The standard notes further state that the correlation relationship used to determine the strength of concrete provides the required standard level of safety, where 90% of the strength values are expected to be higher than the value determined from the relationship. The base curves provided in the standard are defined as the lower envelopes of the relationship between the non-destructive test results (in this case reflection number L) and the concrete strength f_c,cyl_. It should be noted that the standard base curves were determined for f_c,cube_ values determined on 150 mm cube specimens tested after 28 days of curing under laboratory conditions.

Since the study obtained average L_mv_ reflection numbers in a range specified as 43.8–48.0, a standard base curve [59] from Equation (6) was adopted for scaling:(6)$$f_{L}=1.37\times L-\;34.5$$

The evaluation of concrete strength consists in shifting the basic base curve (Equation (5)) to an appropriate level, determined by core drilling and non-destructive testing. The value of the shift of the basic Δ*f* of the base curve depends on the mean value of the differences δf_m__(__n__)_ and the factor k_1_ associated with the number of measurements and is calculated from Equation (7).
(7)$$\Delta f=\delta f_{m\left(n\right)}-k_{1}\times s$$
where:$\delta f_{m\left(n\right)}$—the mean value of the difference of the compressive strength determined on the core bore samples and the strength determined from the base curve (Equation (6)),*s*—standard deviation of strength differences $\delta f_{m\left(n\right)}$,$k_{1}$—coefficient taken from the standard table, for *n* ≥ 9.

From Equation (6), the compressive strength of concrete (corresponding to the strength of concrete on 150 mm cubes) can be calculated anywhere in the structure, then the characteristic compressive strength of concrete dependent on the number of measurements can be determined. See Equation (8):(8)$$f_{is,R}=f_{R}+\Delta f$$

The classification of concrete into a given strength class [59] is based on calculation of the strength from the scaled base curve described by Equation (8) and divided by a factor β_cc_(t) that takes into account the age of the concrete after 28 days of curing [51] (Equations (9) and (10)):(9)$$f_{cm}\left(t\right)=\;\beta_{cc}\left(t\right)\times \;f_{cm}$$
and:(10)$$\beta_{cc}=exp\left[s_{c}\times \left(1-\sqrt{\frac{28}{t}}\right)\right]$$
where:*t*—age of structure (days),*s_c_*—cement class.

In the standard, the method of determining the shift parameter ∆*f* depending on the standard deviation s of the differences $\delta f_{m\left(n\right)}$ and the parameter (statistic) $k_{1}$. remains a matter of debate. In this procedure, the values of the parameter $k_{1}$. are assumed to correspond to the coefficients used in consistency criteria based on the operational characteristic function method, analogous to [60], which should not be used for correlation analyses. In place of the proposed baseline curve shift parameter, it would be more reasonable to use generally accepted principles of mathematical statistics and a shift parameter that depends on the statistics of the assumed distribution. Current recommendations assume a large number of samples *n* ≥ 9 as the basis for scaling the *f_R_*-_R_. relationship. For approximate scaling, the number of pairs of results *n* ≥ 3 may already be sufficient, similar to national (Polish) regulations applied to the sclerometric and ultrasound methods [72,73]. In the analyzed case, the number of results from core samples is *n* = 8, so on the basis of national experience [71,72,73] the authors proposed a modification of the standard method using as *k*_1_ as the value of statistics recommended in [75]. In the calculation, the confidence level *γ* = 0.75 and *n* = 8, and the statistic value read from [75] is *k*_1_ = 3.27. The calculation of the base curves is shown in Table 7.

According to Equation (7), the value of the parameter Δ*f* of the base curve is ∆*f =* 4.9 − 3.27∙3.0 = −4.9 MPa, and the corrected equation of the base curve determined by Equation (5) will finally take the form (11):
*f_is,L_* = *f_L_* + Δ*f* = 1.73 *L* − 34.5 − 4.9 = 1.73 *L* − 39.4(11)

Equation (11) is valid only in the interval of rebound numbers 44.8 − 2 = 42.8 < L < 46.8 − 2 = 44.8, because a modified base curve has been defined for this interval.

The purpose of the analyses was to determine the characteristic strength of concrete in the structure using the indirect method based on the modified base curve (Figure 14), so the calculated values of concrete compressive strength for the base curve, modified base curve, and core specimen methods are shown in Table 8.

The characteristic strength of concrete according to [59] was evaluated using a modified base curve. The criterion for the compliance of the strength of the inspected concrete with the required characteristic strength with the number of specimens *n* < 15 is presented as follows:1st criterion: $f_{ck,is}=f_{cm\;\left(8\right),is,cube}-4\mathrm{MPa}=39.6-4\;\mathrm{MPa}=35.6\;\mathrm{MPa}$2nd criterion: $f_{ck,is}=f_{c\;lowest,cube}+4\mathrm{MPa}=38.1+4\;\mathrm{MPa}=42.1\;\mathrm{MPa}$

The characteristic strength of the concrete is 35.6 MPa. After assuming that class N cement was used to make the concrete and the age of the concrete was more than 50 years, the value of the coefficient considering the age of the concrete β_cc_(t) = 1.438 was determined from Equations (9) and (10); hence, the strength of the concrete was 35.6 MPa/1.438 = 24.8 MPa.

When using the modification of the method, the formal conditions of the core boreholes should be checked. The coefficient of variation of the reflection number at the locations of the drilled wells is v_L_ = 2.0% < 15% (Table 7), and the quotient of the average compressive strength obtained from the curve and the wells is equal to (39.6/40.5 L) − l00% = 2% < 15%. The wells can be considered authoritative and fully representative. The obtained value of relative standard deviation was v_k_ = 2.2%, so the developed modified base curve can be considered the basic scaling curve for the case in question and the small sample size is *n* = 8.

## 4. Discussion

### 4.1. Deflection Measurement Discussion

In many cases in construction practice, it may not be possible to carry out a precise assessment of the technical condition of buildings and their components using traditional, commonly used methods.

The authors have been observing cable post-tensioned concrete girders for several years, cyclically evaluating the material and geometric changes occurring in them. In the analyzed cases, the girder maximum deflection increment Δ*f* in the middle of the span, monitored from 2009 to 2020 in Hall A was 25 mm (about 23% of the maximum deflection) and in hall B, 32 mm (about 33% of the maximum deflection). These values are high for this type of structure because of the lack of concerning amount of girder deformation since the beginning of the hall exploitation. The first measured deflections occurred in 2009 and these values were taken as a reference for later measurements and analyses. According to the authors, because of the age of the elements and incomplete measurement data, the constant measured deflection of the girders in the middle of the span should not exceed 35% of the permissible value. One of the aims of the research was to monitor the changes of displacement magnitude in the same points with changing sign, in similar conditions of the internal and external climate. The change of the ordinates of the measuring points exceeding the permissible values will indicate the overloading of the girder structure, the settling of supports, or the change of tension force in cables. It will become the reason for the detailed examination of this particular element (not all of them) after its temporary protection.

Of course, such tests should not only concern roof structures, but also all other elements which are responsible for their proper support and ensure the spatial stiffness of the objects and their individual structures, and thus affect their deflections.

The hidden defects of built materials as well as the ageing processes will determine their current technical condition, which will constantly change over time to their disadvantage. Periodic examinations, carried out on an ongoing basis, allow for the continued periodic use of the facilities, but at some point they should indicate the approaching end of their service life. Assessment of the parameters of the materials which make up these structures on its own will not be sufficient here, as it will be difficult or even impossible to reach all their locations. It will also not be possible here to assess the tension of cables due to their inaccessibility or to reliably assess the degree and extent of corrosion of steel cable strands (apart from single random tests) due to their number, as there are dozens or even hundreds of them in such halls. In the absence of developed testing methods for this type of structures, which are estimated to be even more than 60 years old, it is necessary to use those that monitor the maintenance of their geometric shapes not only in comparison with the permissible deformations but also through the analysis of these changes over time.

### 4.2. Concrete Test Discussion

The obtained values of the characteristic compressive strength were compared with the values of the characteristic strength calculated on the basis of the guidelines contained in Annex D of PN-EN 1990 [75] concerning research-assisted design.

According to Annex D, the procedure according to D.7.2 and D.7.3 can be directly applied to estimate the characteristic values. It is recommended that the estimation of test results should be carried out on the basis of statistical methods using existing (statistical) information about the type of distribution used and the associated parameters. Equation (12) can be used to estimate the characteristic value according to EN 1990 [75]:
*f_ck_* = *f_cm_* (1 − *k_n_* × *v_k_*)(12)
where:*v_k_*—coefficient of variation of the sample; *v_k_* = *s/f_cm_**k_n_*—a factor that depends on the sample size and confidence level (at the minimum confidence level *γ* = 0.75 and *n* = 8 → *k_n_* = 5.07).

The resulting characteristic compressive strength of concrete according to Equation (12) will be *f_ck_* = 35.9 MPa. The characteristic value of the compressive strength of concrete can also be determined according to EN 1990 [75] in Annex D and the theoretical value of the standard deviation *σ* = 4.86 MPa, calculated from the relationship *f_cm_ = f_ck_ +* 8 according to [51].

With an assumed a priori standard deviation of 4.86 MPa, the standard deviation estimator *s* can be obtained using the Mellin function (Equation (13)):(13)$$s=\sigma \times \left(\sqrt{\frac{2}{n}}\times \frac{\Gamma \left(\frac{n-1}{2}\right)}{\Gamma \left(\frac{n}{2}\right)}\right)$$
where:$\Gamma \left(\ldots \right)$—gamma function value,*n*—sample size,*σ*—population standard deviation.

To determine the characteristic compressive strength of concrete according to [75], we use Equation (14):(14)$$f_{ck}=f_{cm}^{}-k_{n}\times \sigma \times \left(\sqrt{\frac{2}{n}}\times \frac{\Gamma \left(\frac{n-1}{2}\right)}{\Gamma \left(\frac{n}{2}\right)}\right)$$

Under the assumption regarding the knowledge of standard deviation, the value of coefficient will be *k_n_* = 3.27, while the characteristic value of compressive strength according to Equation (14) for core samples will be *f_ck_* = 33.0 MPa.

After considering the age parameter of concrete, the characteristic compressive strength for the core test is, 35.9/1.438 = 29.4 MPa and 33.0/1.438 = 27.0 MPa respectively.

To determine the differences in the characteristic compressive strength values obtained by different methods, the values from each destructive and non-destructive method are summarized in a graph (Figure 15). In the characteristic compressive strength values obtained by non-destructive methods, sclerometric and ultrasonic, a coefficient related to the age of concrete is also included (Equations (9) and (10)).

The typical value (of the compressive strength characteristic values, understood as (*f_ck(5)_* − *s; f_ck(5)_* − *s*), is in the range of 20.3–28.4 MPa, and the relative variation of the compressive strength characteristic value measured by the coefficient of variation is 17% > 15%, which indicates the average variation of the compressive strength characteristic results obtained using different methods.

On the basis of Figure 15, it can be stated that, based on the recommendations [71], the characteristic values of compressive strength obtained using non-destructive methods, i.e., sclerometric and ultrasonic, are correct and smaller than those obtained using the destructive method or the author’s mean method. It should be stressed that in non-destructive methods these are characteristic values of compressive strength, i.e., concrete class determined according to the standard used during the construction of cable concrete girders under consideration, taking into account the age of concrete. The values obtained indicate the correctness of the standard compliance criteria recommended at that time and the possibility their use in the evaluation of concrete in the structure using non-destructive methods.

Based on the results of the tests (Figure 15) of the characteristic compressive strength of concrete of the analyzed structural members using certain methods, i.e., sclerometric, ultrasonic, and indirect with the use of the author’s proposal to modify the base curve using the procedures recommended in [75], the obtained concrete class is C15/20. In the case of applying the destructive method (core drilling), the concrete in the analyzed girders can be classified to class C20/25. Moreover, according to the author’s proposed method, the concrete is on the border between classes C15/20 and C20/25. The design class of concrete for the analyzed girders is C25/30, so the requirement for the design class of concrete is not fulfilled at present. Evaluating the concrete according to the standard recommendations [59], the concrete in the analyzed structure is one (destructive method) or two classes (non-destructive methods) lower than the class assumed for the design concrete.

The choice of test method should be determined by the reliability of the strength estimation. In the presented situation for cable concrete girders, it is very difficult to decide where to take the samples, because they are obtained from where access is the easiest, there are no collisions with the existing reinforcement (in this case tendons and their anchorage,) and also soft reinforcement, or from where the impediments to the use of the object are the least, hence the rationale for using primarily non-destructive methods. Such situations are often the result of a lack of prior visual inspection of the structure and assessment of potential obstructions. In the case under analysis, a total of *n* = 8 samples were taken; however, *n* = 2 samples were taken for one element. This number is less than the *n* = 3 recommended in the standard, which would allow the determination of the characteristic compressive strength of concrete and the subsequent qualification of concrete to a given class. According to [75], non-destructive methods are not an alternative to destructive testing, but only a supplement in the case of a limited number of boreholes. However, in the case of prestressed or post-tensioned concrete structures, it is difficult to take core samples, so only non-destructive methods and the application of appropriate compliance criteria should be considered for the proper classification of concrete.

Sampling of *n* = 3 is allowed by the standard and is justified when the tests concern a single element or a small fragment of the structure. When using the direct method, a minimum of 4–6 core holes should be made in single elements, and when the assessment concerns several elements (substrings, beams, and slabs or columns) then a minimum of 9–18 holes should be made and a combination of indirect methods of testing should be used. Analyzing the obtained values of coefficients of variation for compressive strength, it can be stated that they are less than 15%, which indicates a very good homogeneity of compressive strength of hardened concrete. Very good homogeneity in compression is also evidenced by the obtained values of the homogeneity coefficient for non-destructive methods: sclerometric and ultrasonic.

The calculation procedures recommended in the standards proposed in this paper can enable the proper classification of concrete. It should be noted that using the criteria given in the standard [60], which apply to manufactured concrete, one can obtain dangerously inflated concrete strengths in a structure. When a small sample size of *n* < 15 is available, the recommendations of the standard [59] should be used. Using the compatibility criteria given in [59], the cores taken from the structure allow the characteristic strength of the concrete to be determined, which must be taken into account when determining the safety of the structure. In this case, the characteristic strength of the concrete in the structure must be “reduced” to the concrete strength on the 28th day of concrete curing in laboratory conditions by dividing it by a factor of 0.85 and taking into account the age of the concrete.

Implementation of European standards in the field of testing procedures (PN-EN 12504 series [61,62,63,64]) and concrete qualifications [59] has significantly clarified the situation in the field of the diagnostics of existing reinforced concrete structures. The gap in national (Polish) regulations concerning the methodology of direct destructive testing has been filled and at the same time new procedures for performing indirect non-destructive tests have been given. Both the standard for manufactured concrete [60] and the standard for testing concrete in a structure [59] as well as other subject standards (for non-destructive testing) clearly emphasize that non-destructive methods cannot be used without appropriate scaling on core samples taken from the structure. This highlights the problem of concrete evaluation in post-tensioned or prestressed concrete structures, where it is very difficult to take a sample of a statistically significant number and to use only non-destructive methods. When determining the number of samples, one should not be guided by economic criteria but by the needs of the tests performed and the reliability of the results obtained. Usually, limiting the number of samples to the absolute minimum (*n* = 3) results in the necessity of additional tests. To exclude doubts arising at the stage of drawing conclusions, in situ tests should be attended by representatives of the interested parties, whose task it is to both confirm the location of the test points and visually evaluate the samples and, if possible, observe the test procedures.

## 5. Conclusions

A reliable condition assessment of cable post-tensioned concrete girder structures is very difficult to perform in their “natural” operating environment. Nevertheless, it is necessary to maintain such structures in a safe technical condition. A serious problem is the age of these structures, sometimes exceeding 60 years. The danger associated with this may appear suddenly and uncontrollably, leading to the collapse of entire facilities without any warning signals. A big problem is the assessment of the technical condition of covered and inaccessible elements of girders, mainly steel tendons and their anchorages. Therefore, people carrying out such investigations are required not only to have advanced theoretical knowledge, but also practical experience, expertise, and above all common sense. Summarizing the content of the paper, it can be concluded that:monitoring of existing building structures containing prestressed structures is essential for economic reasons, safety of use, durability, and determination of the moment until which they can be used;the methods proposed in this paper are useful or even necessary to determine and monitor the current technical condition of such objects “in situ”;the method of testing deformation and displacements of prestressed elements is the basic method, which makes it possible to assess the technical condition of prestressed elements, whereas non-destructive strength testing may be an auxiliary and supplementary method.

Those persons responsible for the exploitation of such objects should take special care to maintain the internal microclimate at a stable level and prevent any adverse phenomena (e.g., leaks in roof coverings) which could accelerate corrosion, mainly of invisible steel structures “covered” in the girder body.

## Figures and Tables

**Figure 1 materials-14-01503-f001:**
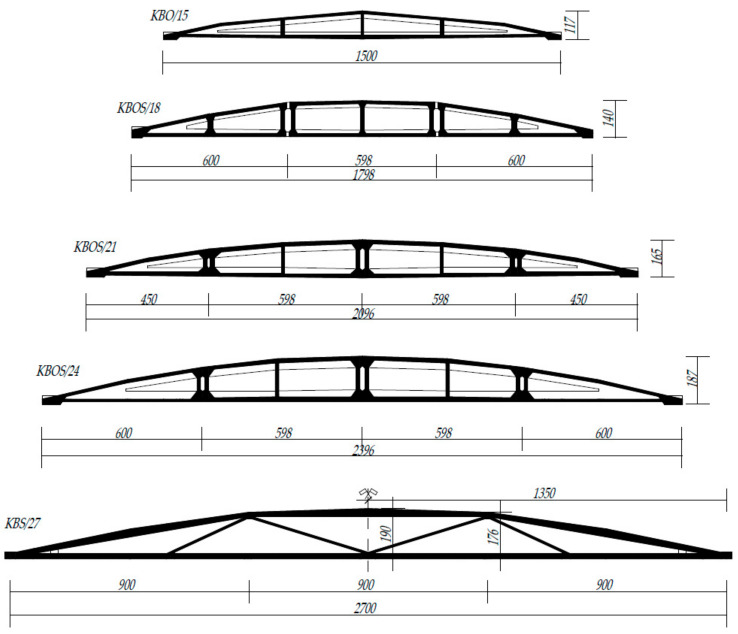
Form and dimensions of example cable post-tensioned concrete girders.

**Figure 2 materials-14-01503-f002:**
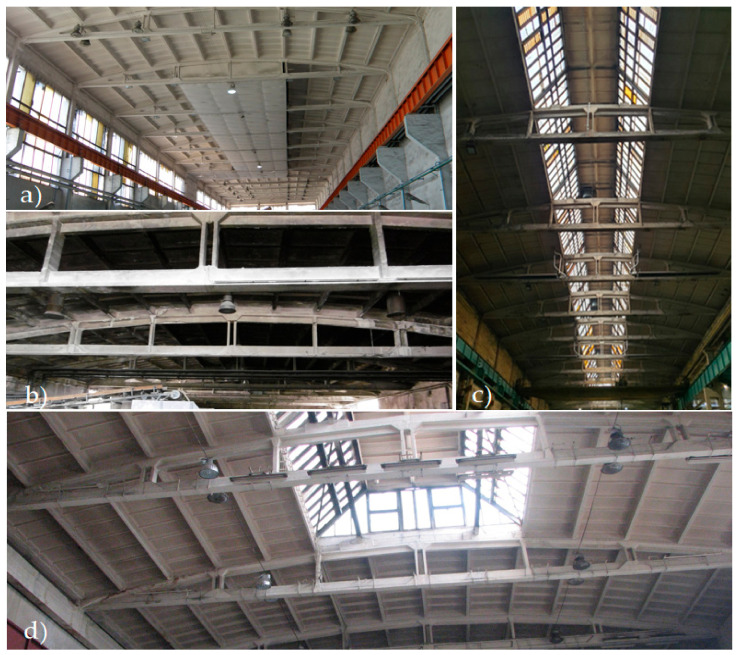
Factory buildings with a large number of cable post-tensioned concrete girders: (**a**) production and assembly hall; (**b**,**c**) production hall; (**d**) storage hall for powdered materials.

**Figure 3 materials-14-01503-f003:**
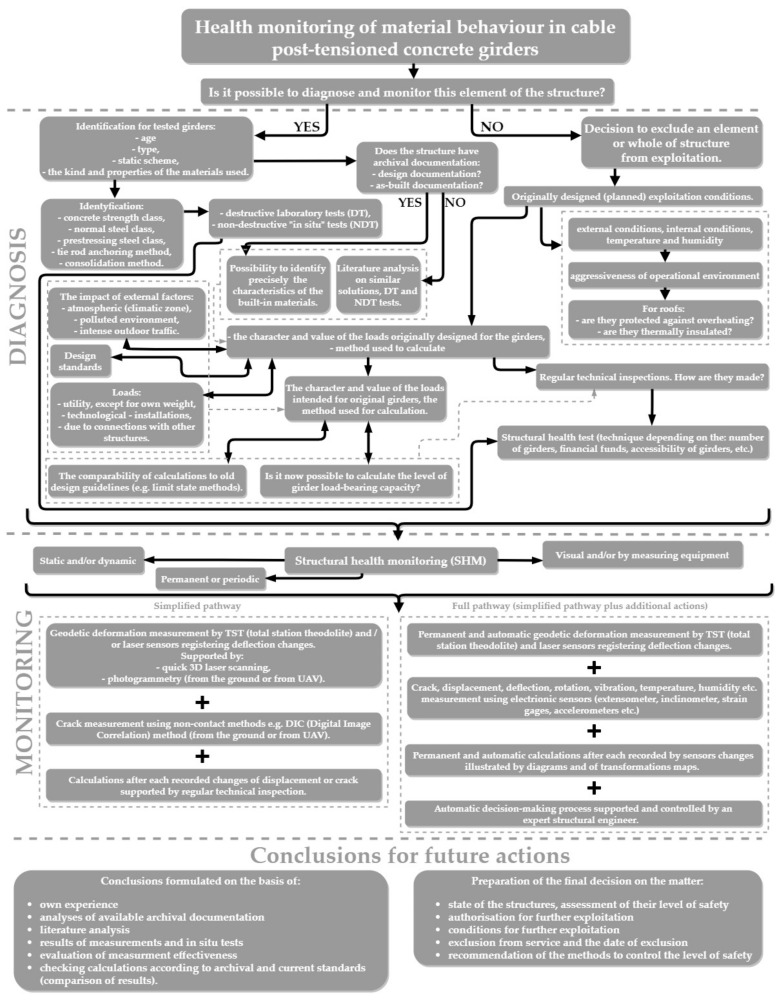
Flowchart for quick and reliable material health monitoring and structure behaviour.

**Figure 4 materials-14-01503-f004:**
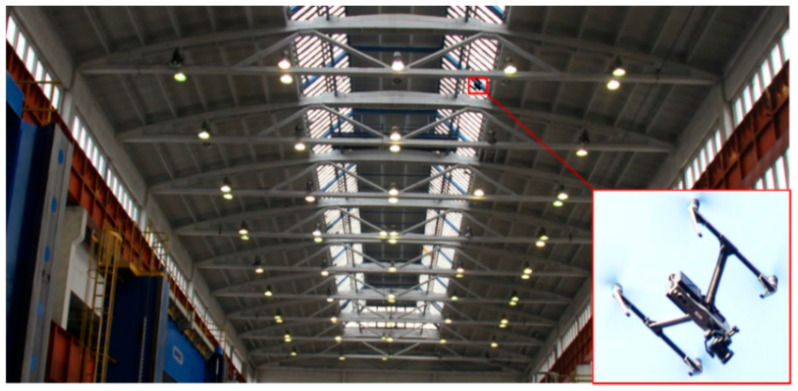
Visual inspection using an unmanned aerial vehicle (UAV).

**Figure 5 materials-14-01503-f005:**
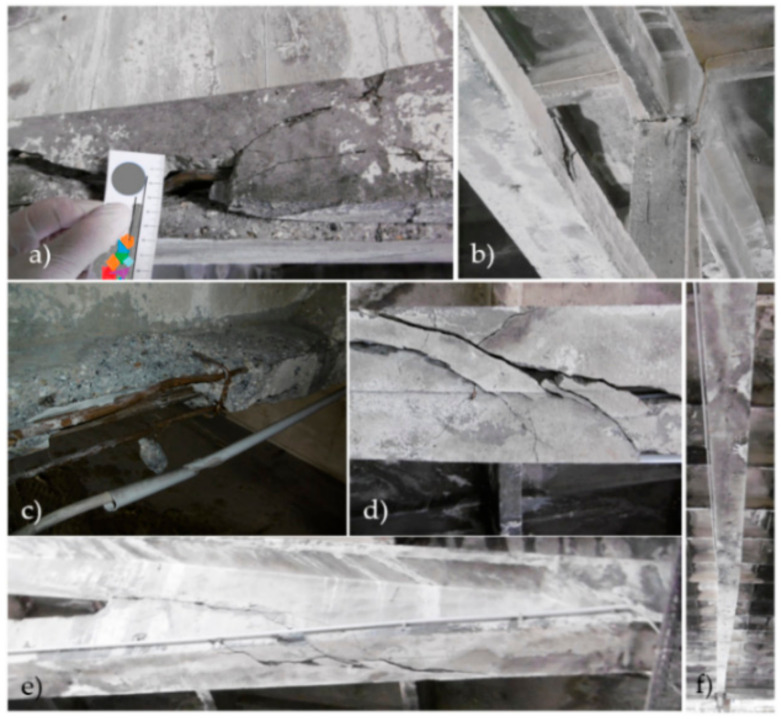
Examples of cable post-tensioned concrete girder damage in the Hall A inventoried from: (**a**) a basket lift; (**b**–**f**) using UAV**.**

**Figure 6 materials-14-01503-f006:**
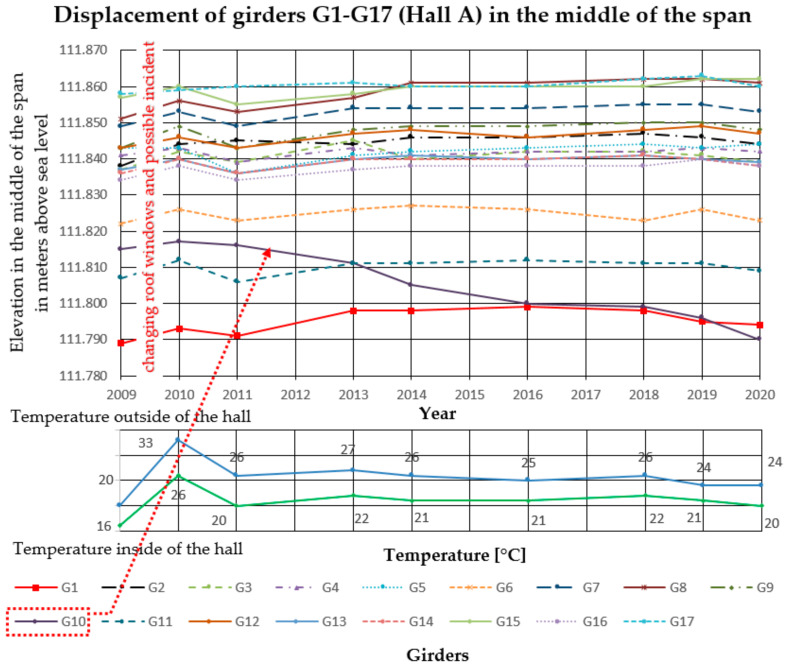
Displacement of girders in the middle of the span in Hall A. Visible significant change in the deflections of G10 girder detail shown in Figure 5.

**Figure 7 materials-14-01503-f007:**
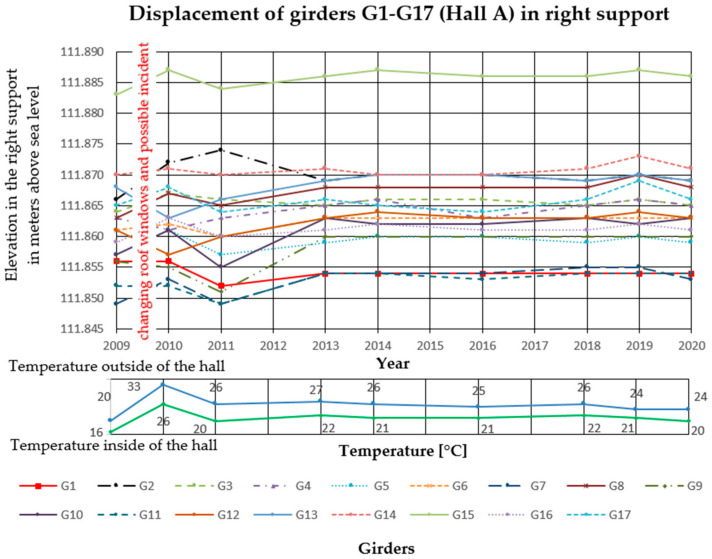
Displacement of the right side of the girders in Hall A.

**Figure 8 materials-14-01503-f008:**
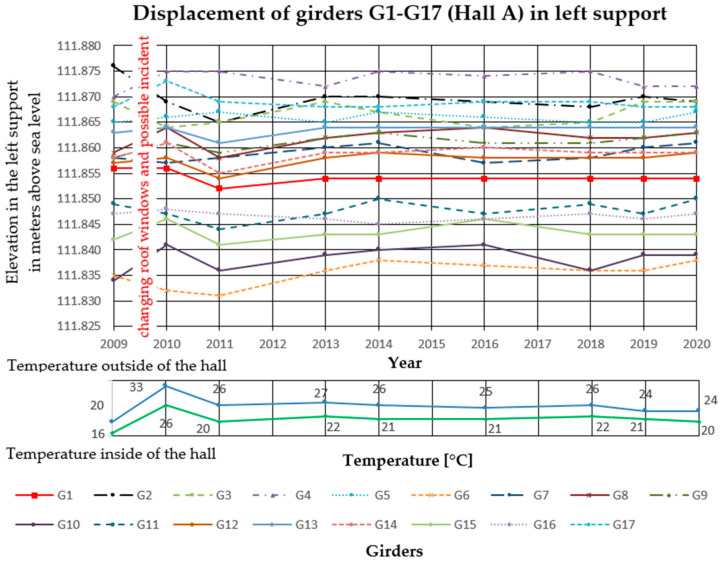
Displacement of the left site of the girders in Hall A.

**Figure 9 materials-14-01503-f009:**
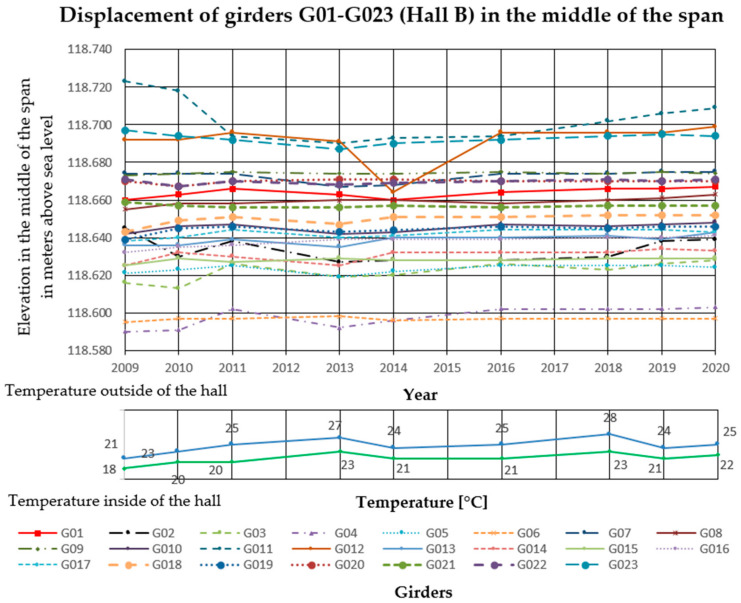
Displacement of the girders in the middle of the span in Hall B.

**Figure 10 materials-14-01503-f010:**
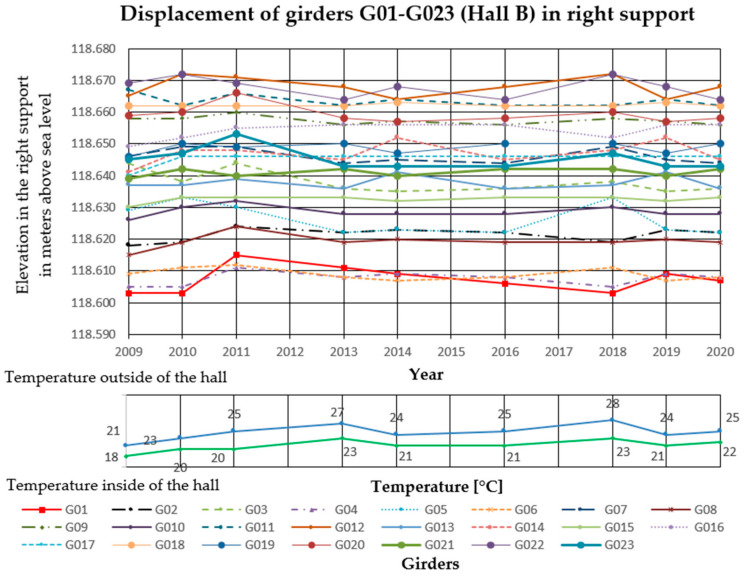
Displacement of the right side of the girders in Hall B.

**Figure 11 materials-14-01503-f011:**
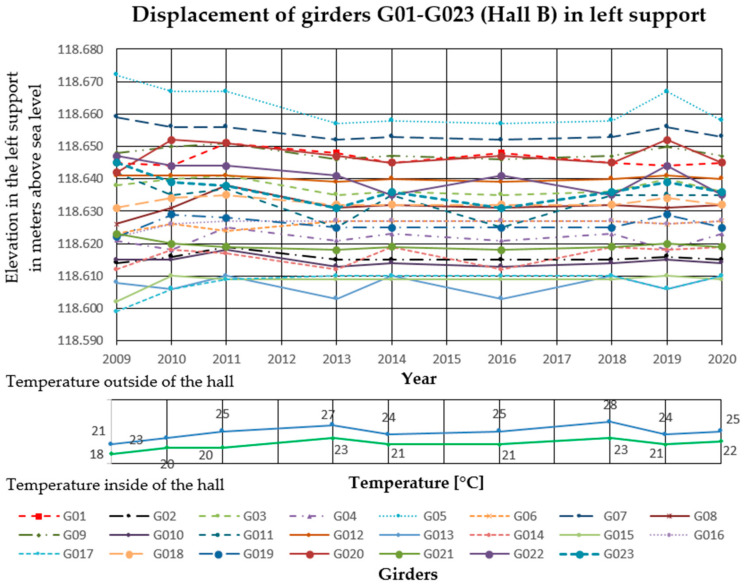
Displacement of the left side of the girders in Hall B.

**Figure 12 materials-14-01503-f012:**
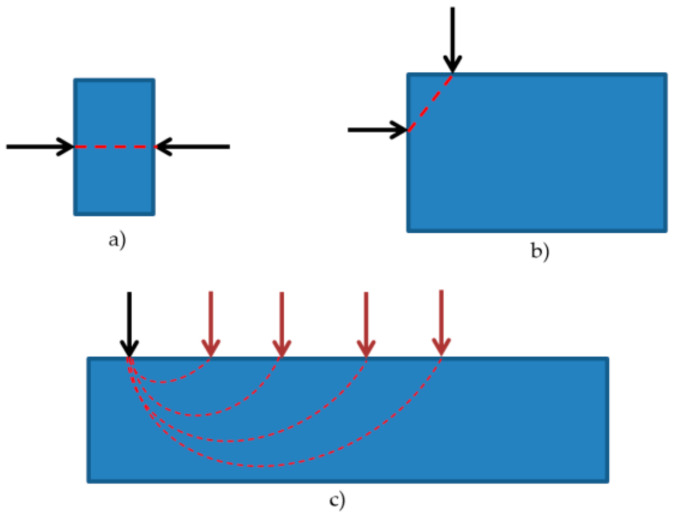
Wave velocity measurement—ultrasonic testing schema: (**a**) crosswise method; (**b**) semi-direct method; (**c**) surface method.

**Figure 13 materials-14-01503-f013:**
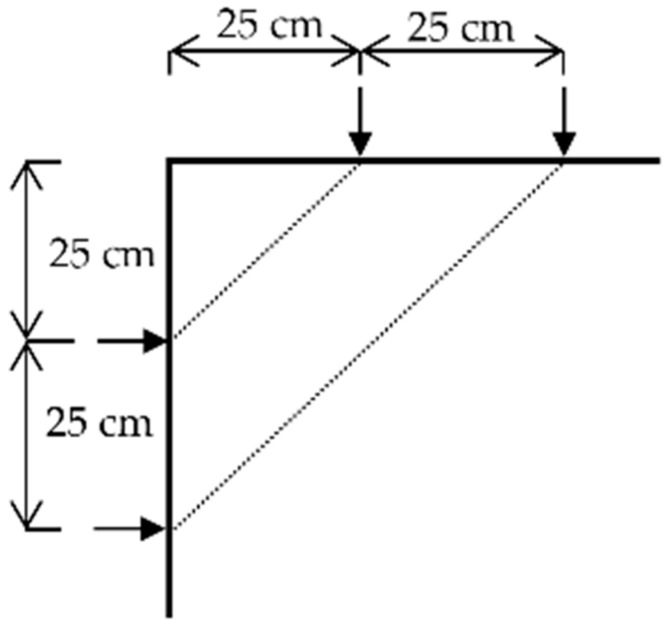
Scheme of transducer positioning in ultrasonic testing.

**Figure 14 materials-14-01503-f014:**
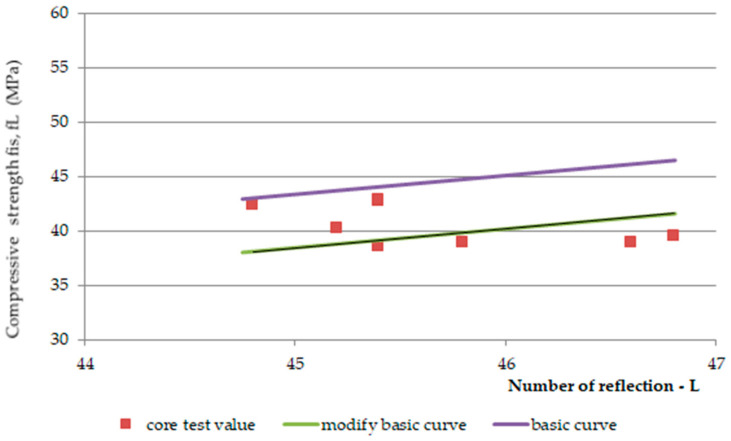
Test results according to the indirect method taking into account the results of destructive and non-destructive tests.

**Figure 15 materials-14-01503-f015:**
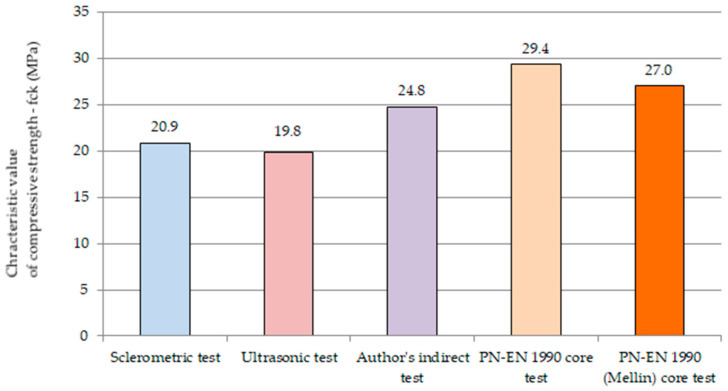
Characteristic values of compressive strength obtained using different methods.

**Table 1 materials-14-01503-t001:** Classification of non-destructive (NDT) concrete testing methods.

Group of Methods	Type of Method	Test Instruments	Tested Parameter	Specified Concrete Properties	Comments	Advantages	Disadvantages
Sclerometric methods	Impact methods	Poldi hammer, HPS hammer	imprint depth and diameter	hardness, compressive strength	historical methods, rarely used today	fixed interaction energy	necessity of measuring the imprint diameter increases the testing time
	Rebound methods	Sclerometers (Schmidt hammer L, N, M, PT type)	number of rebounds of the impact mass acting on the component under test with constant energy	hardness, compressive strength	method often used today	fixed interaction energy, measurement speed	influence of concrete age on reflection number
Acoustic methods	Ultrasonic method	ultrasonic pulse velocity testers, material samplers	speed of the ultrasonic wave that spreads in the test material	concrete compressive strength, concrete homogeneity, defect detection (defectoscopy)	two ultrasonic heads: transmitting and receiving	totally non-destructive method	influence of reinforcement on ultrasonic wave velocity measurement, discrepancies in results
	Ultrasonic tomography method	ultrasonic tomographs	measurement of the propagation of elastic waves (ultrasonic) induced by a multi-head antenna	detection of various types of material imperfections, with one-sided access	multiple transmitter/ receiver heads	totally non-destructive method, high accuracy	high work intensity of the tests, discrepancies in results
	AE Acoustic Emission	AE instruments	energy and amplitude of the acoustic wave generated in a component as a result of its loading or deformation	defectoscopy	the method is mainly used for steel elements, less frequently for concrete elements	totally non-destructive method, high accuracy	high work intensity of the tests, discrepancies in results
	Echo method	Specialised measuring equipment	transition time of the pulse reflected from the opposite surface	defectoscopy	one transmitter/ receiver head	totally non-destructive method, high accuracy	high work intensity of the tests, discrepancies in results
	Hammer method	Specialised measuring equipment	velocity of the ultrasonic wave excited in the component under test	defectoscopy	acoustic impulse is generated by hitting the surface of the test piece using a hammer	totally non-destructive method, high accuracy	high work intensity of the tests, discrepancies in results
Quasi-destructive metod	Methods of pulling out an anchor or bolt anchored in concrete	Equipment: - pull off, - pull out, - lock test	pull-out force	compressive strength	there is local damage of approx. 5 cm depth from the surveyed surface	possibility of fixing anchors in concrete as well as in hardened concrete	high work intensity of fixing the anchors in hardened concrete
	Borehole method	Drill rig with core bit, testing machine	destructive force of specimens recovered from structures by means of boreholes	compressive strength	need to collect concrete samples	high accuracy	high work intensity of the sampling

**Table 2 materials-14-01503-t002:** Sclerometric test results (girder G11, Hall A).

No.	Testing Angle *α*	Mean Rebound Value *L_mv_*	Equivalent Mean Rebound Value *L_mv(_**_α_**_–0)_*	Strength from the Curve *f_ci_*	Corrected Strength *f_ci_′* = 0.6 · *f_ci_*
1	0	46.6	46.6	63.1	37.9
2	0	45.4	45.4	59.1	35.6
3	0	45.2	45.2	58.4	35.1
4	0	45.8	45.8	60.4	36.3
5	0	45.4	45.4	59.1	35.5
6	0	46.8	46.8	63.8	38.3
7	0	44.8	44.8	57.1	34.3
8	0	45.4	45.4	59.1	35.5
9	0	45.8	45.8	60.4	36.3
10	+90°	44.4	45.8	60.4	36.3
11	+90°	44.6	48.0	67.9	40.8
12	+90°	40.6	43.8	53.9	32.4
13	0	45.2	45.2	58.4	35.1
14	0	45.2	45.2	58.4	35.1
15	0	45.4	45.4	59.1	35.5
Average compressive strength			*f_c mean_* = 36.0 MPa		
Standard deviation			*s* = 1.92 MPa		
Relative standard deviation			*υ**_k_* = 5.3%		

Tested component: cable post-tensioned concrete roof girder. Age of concrete: more than 1000 days, correction factor 0.6. Measuring device: N-type Schmidt sclerometer. Test angle α: 0° (horizontal test), +90° (test from bottom site).

**Table 3 materials-14-01503-t003:** The homogeneity of hardened concrete from sclerometer test (girders G10 & G11—Hall A, girders G05 & G018—Hall B).

Girder	Homogeneity of Hardened Concrete According to [72]
G10—Hall A	*k* = 0.90 > 0.8; Very good
G11—Hall A	*k* = 0.91 > 0.8; Very good
G05—Hall B	*k* = 0.89 > 0.8; Very good
G018—Hall B	*k* = 0.92 > 0.8; Very good

**Table 4 materials-14-01503-t004:** Ultrasonic test results (girder G11, Hall A).

No.	Length of the Measurement Path *s* (mm)	Pulse Transition Time *t* (μs)	Ultrasound Wave Speed *v_L_* (km/s)	Concrete Compressive Strength *f_c_* (Mpa)
1	210	48.1	4.4	32.6
2	210	47.2	4.4	34.7
3	205	46.9	4.4	32.8
4	200	46.6	4.3	30.9
5	205	46.7	4.4	33.2
6	205	46.1	4.4	34.6
7	210	46.1	4.6	37.3
8	210	46.3	4.5	36.8
9	205	46.4	4.4	33.9
10	210	46.3	4.5	36.8
11	205	46.2	4.4	34.4
12	210	46.5	4.5	36.3
13	200	46.9	4.3	30.3
14	205	46.7	4.4	33.2
15	210	46.9	4.5	35.4
Average compressive strength			*f_c mean_* = 42.0 Mpa	
Standard deviation			*s* = 2.11 MPa	
Relative standard deviation			*υ**_k_* = 6.2%	

**Table 5 materials-14-01503-t005:** The homogeneity of hardened concrete from ultrasonic test (girders G10 & G11—Hall A, girders G05 & G018—Hall B).

Girder	Homogeneity of Hardened Concrete According	
	to [73]	to [74]
G10—Hall A	*k* = 0.89 > 0.8; Very good	*v* = 4.0 m/s ∈ (3.5–4.5) m/s; Good
G11—Hall A	*k* = 0.90 > 0.8; Very good	*v* = 4.4 m/s ∈ (3.5–4.5) m/s; Good
G05—Hall B	*k* = 0.87 > 0.8; Very good	*v* = 3.8 m/s ∈ (3.5–4.5) m/s; Good
G018—Hall B	*k* = 0.92 > 0.8; Very good	*v* = 4.4 m/s ∈ (3.5–4.5) m/s; Good

**Table 6 materials-14-01503-t006:** Determining the strength of structural concrete (girders G10 & G11—Hall A, girders G05 & G018—Hall B).

Girder/Sample no.	Destructive Force *F* (kN)	Concrete Compressive Strength *f_c_* (MPa)
G10—Hall A/001	254.0	33.1
G10—Hall A/002	248.0	32.7
G11—Hall A/003	267.0	34.2
G11—Hall A/004	260.0	33.1
G05—Hall B/005	279.0	36.3
G05—Hall B/006	264.0	33.6
G018—Hall B/007	283.0	36.0
G018—Hall B/008	286.6	36.4
Average compressive strength of concrete		*f_cm_* = 34.4 MPa
Standard deviation		*s* = 1.6 MPa
Relative standard deviation		*υ**_k_* = 4.5%

**Table 7 materials-14-01503-t007:** Calculations for the base curve offset parameter.

Girder/Sample no.	Compressive Strength		*δf* = *f_is_* − *f_L_* (MPa)
	Core with Correction Value 0.85 *f_is_* (MPa)	Basic Curve—Equation (5) *f_L_* (MPa)	
G10—Hall A/001	38.9	22.8	6.1
G10—Hall A/002	38.5	22.1	6.4
G11—Hall A/003	40.2	24.7	5.1
G11—Hall A/004	38.9	22.8	6.1
G05—Hall B/005	42.7	28.3	3.3
G05—Hall B/006	39.5	23.6	5.6
G018—Hall B/007	42.4	27.8	3.6
G018—Hall B/008	42.8	22.1	3.2
Average value		*δf**_m(_*_8__)_ = 4.9 MPa	
Standard deviation is		s = 1.34 MPa < 3.0 MPa	
and for calculation according to [44]			
standard deviation can be use		s = 3.0 MPa	

**Table 8 materials-14-01503-t008:** Compressive strength obtained from destructive and non-destructive testing.

Girder/Sample No.	Core Vale *f_is_* (MPa)	Compressive Strength Determined from the Curve	
		Basic Curve Equation (6)	Modified Equation (11)
G10—Hall A/001	38.9	46.1	41.2
G10—Hall A/002	38.5	44.0	39.1
G11—Hall A/003	40.2	43.7	38.8
G11—Hall A/004	38.9	44.7	39.8
G05—Hall B/005	42.7	44.0	39.1
G05—Hall B/006	39.5	46.5	41.6
G018—Hall B/007	42.4	43.0	38.1
G018—Hall B/008	42.8	44.0	39.1
Average value—*f_c_*_(8) is, cube_ (MPa)	40.5	44.5	39.6
Lowest value—*f_clowest,_*_cube_ (MPa)	38.5	43.0	38.1
Standard deviation s (MPa)	1.8	1.2	1.2
3.0 MPa, and for calculation according to [44] standard deviation is *s* = 3.0 MPa			

## Data Availability

Data sharing is not applicable.

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
