# Peer review of "Elements of Pathway for Quick and Reliable Health Monitoring of Concrete Behavior in Cable Post-Tensioned Concrete Girders"

_materials, 2021, doi:10.3390/ma14061503_

Round 1

Reviewer 1 Report

Dear Editor,

The topic of the paper is interesting and suits the Journal of Materials. However, a minor revision is required before this manuscript is qualified to be published in this prestigious journal. The manuscript is needed to be revised grammatically. The authors are required to check the whole manuscript with a grammar specialist as it has several grammatical errors. Only after revising the manuscript based on the comments, the paper is suggested to be published in MDPI. Further information on various issues identified in the manuscript appears below:

  1. The authors have done a great job on the literature review. However, the introduction needs more attention. More information on the new materials:

"Connections and structural applications of fibre reinforced polymer composites for civil infrastructure in aggressive environments." Composites Part B: Engineering 164 (2019): 129-143.

"Fracture Properties Evaluation of Cellulose Nanocrystals Cement Paste." Materials 13, no. 11 (2020): 2507.

  1. Please provide more detailed reasoning behind the behaviors. The details should include rigid numbers or percentages. Please add more theoretical discussion.
  2. Add error bars to your figures where applicable.
  3. Please indicate how many samples for each experiment have been used.
  4. Please describe the process of each experiment. Also indicate the model of each tool that is used in the experiment. What is the accuracy of each test? Please explain them accurately.

Author Response

The authors would like to thank the reviewers for agreeing to review our manuscript and particularly for very useful comments. The corrections are highlighted yellow in the text.

Reviewer 2 Report

Thanks to the authors for their study on a pathway for quick and reliable health monitoring of material behavior in cable post-tensioned concrete girders. The following comments need to be addressed:

  • The Abstract looks more like an introduction. Please summarize your study and major findings in the abstract and remove the current unnecessary sentences.
  • You need to create a separate section from Introduction to completely explain your methodology.
  • How do you consider the influence of concrete aging mechanisms like ASR, DEF, or ACR in your pathway for health monitoring? Please explain.
  • Some concrete aging mechanisms like ASR and DEF cause concrete expansion which may reduce the material properties but have positive impact on the post-tensioning effect of elements. How do you take that into consideration? Please explain.
  • For concrete tests you focused on the compressive strength while in many aging cases compressive strength can not solely represent the condition of concrete without performing tensile and flexural tests. In many occasions the degradation of concrete tensile properties is far worse than the compressive properties. You need to find a way to take this into consideration in your procedure.
  • Please summarize the major findings of your study in bullet points in the Conclusion section.

Author Response

(The authors gave the same response as above.)

Reviewer 3 Report

This paper is an interesting research about Cable Post-Tensioned Concrete Girders. 

Some comments are made to clarify this research and the results obtained:

_ An English review of the full text is very necessary.
_(line 23) It is recommended not to add references to other authors in the abstract. 
_Abstract: it is not clear what is the novelty of this research compared to other studies. Please, put in a clear way what is the interest of this paper.
_Keywords: please incorporate more specific words. What words identify this paper?
_Figure 1: Please check text size. Is the text in these figures readable?. It is necessary to do this figure again.
_Figure 2: Please check text size and the text format. 
_It is important to review the structure of the paper: sections and how they are organized. 
_Table 3. Please check the format. 
_line 365: Figure instead of Figured. 
_line 402. 4. Discussion and Conclusions. It is important to separate discussion from conclusions. It is recommended to check other papers published in this journal to see how it is structured. It is very important to separate the conclusions to identify the main contributions that this research presents.
_ References: Please check the format of all references is in the journal template. There are many that do not adjust with the journal indications.

Author Response

(The authors gave the same response as above.)

Reviewer 4 Report

Dear authors,

Thank you for your interesting research. The manuscript is presenting emerging path in the preservation of existing structures. SHM is a valuable part of the structural engineering, very often unreasonably disregarded.

Here are my comments.

Abstract: Delete the references from the abstract

Keywords: update the list

Line 61 – (70 – 50 years) – change the order

64 - update the list. There are recent guidelines published

72 – update the list with STAR literature

72 Specialised NDT tests can be 72 performed by a few research centres, but they are cautious about this type of objects and materials, probably because of the importance of responsibility. – Reformulate or delete the sentence

74 – 76 Unclear sentence, reformulate

81 their performance? Where is the subject in the sentence

86 Reference 17 is not dealing with a repair

92- 96 Unclear and too long sentence. Reformulate

Figure 1 – The figure is confusing and hard to read. Here are the basic remarks:

  • increase the resolution. Hard to read
  • That the structure has archival?? Reformulate
  • the type and size of the loads?? Reformulate
  • protected against heating?? Reformulate
  • is it now possible…?? Comparability to the old calculations??
  • flying drone (UAV) – it is called unmanned aerial vehicle, no flying drone
  • calculations after each recorded changes? Are you sure this is financially feasible?
  • changes maps??
  • Simplified vs full pathway - why is UAV technique removed from full pathway?
  • …to assess the effectiveness?? Explain and reformulate
  • Checking calculations?? Why is the comparison needed? It is an indication, not the mandatory information
  • Please read the remarks carefully. Please improve English because some parts are not understandable

107 Reformulate – hard to read

115 reformulate. The order of the words in the sentence is wrong

  1. dot is missing in the end of the sentence

122 …increased the permanent load? For sure did, show the number of that increase

139 you are using the word calculations too often. From my point of view, this is not a good word. Calculations of what?

144 support > supports

145 elements cooperating with them – bad English

158 – shorten the sentence

163 – reformulate the sentence

164 The ultimate limit state is a situation in which the values of the internal forces due to the most unfavourable load calculation combination are checked. – This is not true; you didn’t mention that the internal forces are compared with the strength of the material

170 – too long sentence

174 I don’t understand the sentence

179 – google some articles in the journals of Applied sciences and Sensors – there is a lot of new articles with UAV and emerging technologies. The technology is not just for concrete

192 reformulate

205 I don’t understand the sentence

Figure 4 – explain in the text the model of the drone and the resolution of the camera

225 why these assumptions? Elaborate!

Table 1. What’s the point?

236 Why is 10% sufficient?

247 or stopped bridge??

250  The alternative method of horizontal measurement under the roof, on the other hand, does not affect the use of the building. What’s the point of the sentence?

265 reformulate the sentence

Table 2 – try to reference the methods. Comments – try to comment on the advantages and disadvantages of every method

Figure 6 – Changing… Bad English. InsiDE – not insite.. Elevation in m?? Modify it to a height above the ground level. Meters cannot be the measure at the y axis – cm or even mm are important. Same applies to the figures 7-11

Why equations 1 and 2 – elaborate!

352 reference the statements

365 Figure – not figured

383 Please provide the data of the ultrasonic tests!

414 For this reason, among others, in the case described in the article, scientific centres, which were approached, refused to conduct research on the structures in question and to issue appropriate opinions because of the high risk of liability. This is maybe the situation in Poland, it is not like this everywhere in the world. Please reformulate.

416 unclear sentence

435 Specify it! It’s not all of the owners in the world

439 Forgot? Please reformulate

445 Unclear sentence

General remarks:

The topic is interesting. However, it lacks proper and comprehensive methods. The case studies are very well chosen. The case studies should be compared to the other case studies globally – not just the ones from Poland. The situation about the pos-tensioned concrete girders is similar worldwide, and there is a lot more literature about the topic. The title is “Pathway for Quick and Reliable Health Monitoring of Material Behaviour…”. There is nothing about the steel part in the article. If one wants to see the whole procedure, the title of the article will mislead her. The method explained in Figure 1 is confusing. Probably it is easy to read for the group of the authors, but for the rest it’s complicated.

English is, unfortunately, not so good. There are thousands of mistakes and bad formulation of sentences.

The reference list should be updated. The standards/norms are nor referenced in a good way.

Author Response

(The authors gave the same response as above.)

Round 2

Reviewer 2 Report

Thanks to the authors for their responses to my comments. The authors made a good attempt to respond the comments but it is not to my full expectations especially when it comes to the first two comments. 

Reviewer 3 Report

The work of the authors in improving the paper “Pathway for Quick and Reliable Health Monitoring of Material Behaviour in Cable Post-Tensioned Concrete Girders” is appreciated. The authors have considered many of the reviewer’s recommendations and the improvement of the paper has been important.  

However it is necessary to make the following comments:

_(1)Comments regarding the abstract have not been taken into account in its entirety. 
_(2)A review of the structure of the paper (sections and how they are organized) is suggested. Some changes have been made, but perhaps another revision could be made, especially in the first part.
_(3)Figure 1: This figure is not admissible. It is illegible. Please check text size. Is the text in these figures readable ?. It is necessary to do this figure again. This comment was already made in the first revision. 

Reviewer 4 Report

The comments are in the pdf file.
